# Boosted Radiation Bystander Effect of PSMA-Targeted Gold Nanoparticles in Prostate Cancer Radiosensitization

**DOI:** 10.3390/nano12244440

**Published:** 2022-12-14

**Authors:** Daiki Hara, Wensi Tao, Ryder M. Schmidt, Yu-Ping Yang, Sylvia Daunert, Nesrin Dogan, John Chetley Ford, Alan Pollack, Junwei Shi

**Affiliations:** 1Department of Radiation Oncology, Miller School of Medicine, University of Miami, Miami, FL 33136, USA; 2Department of Biomedical Engineering, College of Engineering, University of Miami, Miami, FL 33146, USA; 3Sylvester Comprehensive Cancer Center, Miller School of Medicine, University of Miami, Miami, FL 33136, USA; 4Department of Biochemistry and Molecular Biology, Miller School of Medicine, University of Miami, Miami, FL 33136, USA

**Keywords:** gold nanoparticle, radiosensitization, radiation-induced bystander effect, prostate LNCaP cancer cells, prostate-specific membrane antigen, active targeting

## Abstract

Metal nanoparticles are effective radiosensitizers that locally enhance radiation doses in targeted cancer cells. Compared with other metal nanoparticles, gold nanoparticles (GNPs) exhibit high biocompatibility, low toxicity, and they increase secondary electron scatter. Herein, we investigated the effects of active-targeting GNPs on the radiation-induced bystander effect (RIBE) in prostate cancer cells. The impact of GNPs on the RIBE presents implications for secondary cancers or spatially fractionated radiotherapy treatments. Anti-prostate-specific membrane antigen (PSMA) antibodies were conjugated with PEGylated GNPs through EDC–NHS chemistry. The media transfer technique was performed to induce the RIBE on the non-irradiated bystander cells. This study focused on the LNCaP cell line, because it can model a wide range of stages relating to prostate cancer progression, including the transition from androgen dependence to castration resistance and bone metastasis. First, LNCaP cells were pretreated with phosphate buffered saline (PBS) or PSMA-targeted GNPs (PGNPs) for 24 h and irradiated with 160 kVp X-rays (0–8 Gy). Following that, the collected culture media were filtered (sterile 0.45 µm polyethersulfone) in order to acquire PBS- and PGNP- conditioned media (CM). Then, PBS- and PGNP-CM were transferred to the bystander cells that were loaded with/without PGNPs. MTT, γ-H2AX, clonogenic assays and reactive oxygen species assessments were performed to compare RIBE responses under different treatments. Compared with 2 Gy-PBS-CM, 8 Gy-PBS-CM demonstrated a much higher RIBE response, thus validating the dose dependence of RIBE in LNCaP cells. Compared with PBS-CM, PGNP-CM exhibited lower cell viability, higher DNA damage, and a smaller survival fraction. In the presence of PBS-CM, bystander cells loaded with PGNPs showed increased cell death compared with cells that did not have PGNPs. These results demonstrate the PGNP-boosted expression and sensitivity of RIBE in prostate cancer cells.

## 1. Introduction

Prostate cancers contributed to about 34,500 deaths in the United States in 2022, according to American Cancer Society’s estimates [1]. As a mainstay of cancer treatment, radiotherapy (RT) is commonly used to offer both definitive and palliative strategies for prostate cancer management [2]. The efficacy of RT in cancer therapy stems from the fact that ionizing radiation can directly and indirectly damage DNA and disrupt the atomic structure of biomolecules in the cellular environment [3]. In recent decades, we have witnessed a development boom in terms of high-precision RT techniques, such as intensity modulated radiotherapy (IMRT) and real-time adaptive MRI-guided radiation; these techniques allow improved dose conformity to the tumor target as well as a decreased dose to adjacent healthy tissues [4]. Nevertheless, due to the similar mass energy absorption properties of both cancer and healthy tissues, physical radiation dose escalation and beam conformality has approached an upper limit with regard to prostate cancer external beam RT. Radiosensitizers, including small molecules, macromolecules, and nanomaterials, are promising agents that offer the means for further tumor dose escalation with improved normal tissue sparing [5].

Due to their high mass energy absorption coefficient relative to soft tissue, nanoparticles of high atomic number (Z) materials (such as: iodine, gadolinium, hafnium, tantalum, tungsten, bismuth) have been implemented to improve the contrast between tumors and healthy tissues, thus enabling tumor-specific radiosensitization with reduced side effects [6]. As a promising high-Z nano-radiosensitizer, gold nanoparticles (GNPs) have lately garnered attention due to their special properties; these include high biocompatibility with low toxicity, and the facile attachment of a variety of biological ligands [7]. The efficiency of GNP radiosensitization has been extensively validated in both in vitro and in vivo scenarios using numerous types of ionizing radiation, including kilovoltage (kV) and megavoltage (MV) photons as well as charged particles [8]. Compared with other nanoparticles, GNPs have been well studied for their efficient radiosensitizing effects, their multitude of mechanisms which allows radiosensitization to be carried out, and their comparatively limited toxicity [9]. Although dose enhancement factors vary based on radiation source, Jones et al. reported Monte Carlo simulated microscopic dose enhancement factors of 80× in 50 kVp photon beams and 9.8× in 6 MV photon beams, up to 100 µm, from the GNP surfaces [10]. Similarly, Lin et al. measured a dose enhancement factor of up to 14× in 150 MeV protons at a distance of 10 µm from the nanoparticle surface [11]. Aside from the incident beam, further optimization studies on GNP shape and size suggest that spherical nanoparticles between 10 nm to 20 nm provide the most optimal increase in secondary electron scatter and they minimize the level of toxicity in normal tissue [12].

Blood circulation pathway/time and the extent of tumor accumulation dictates the level of biodistribution, toxicity, and radiosensitization from GNPs. Polyethylene glycol (PEG) is ubiquitously used to coat GNPs, which significantly reduces nonspecific binding with cells and serum proteins, improves GNPs’ stability and biocompatibility under physiological conditions, and it greatly lengthens the circulation half-life of GNPs in vivo [13]. GNPs passively leak into the tumor interstitium from blood vessels feeding the tumor via the enhanced permeation and retention (EPR) effect; this occurs due to the tumors’ leaky vasculature and poor lymphatic drainage [14]. The EPR effect is the rationale behind the passive targeting approach; however, the efficiency of passive targeting is low, and it is still controversial as to whether the EPR effect is relevant in humans [15]. Therefore, the improved accumulation of GNPs with regard to tumors, beyond reliance on the EPR effect, is critical for GNP-based therapies. GNPs are ideal candidates for conjugating tumor-targeting agents because GNP surface chemistry enables a multitude of chemicals to bind at high densities [16]. Among potential targeting agents for prostate cancer, prostate specific membrane antigen (PSMA) ligands have been employed with great success in preclinical and clinical studies for PET imaging and radionuclide therapies; this is because prostate cancers consistently overexpress PSMA [17,18]. Our previous studies have shown the effectiveness of PSMA-targeted GNPs and their ability to accumulate at higher concentrations and to be retained for longer in prostate tumors [19].

Multiple studies have confirmed the radiosensitization of tumor-targeted GNPs in a multitude of tumor tissues, including prostate cancers [8]. It is worth mentioning that the radiosensitization observed in both in vitro and in vivo studies is often significantly greater than the dose enhancement predicted by Monte Carlo computational models [20]. These suggest that complex chemical and biological components, including enhanced ROS production, change during cell cycle distribution. Moreover, these components also affect the overall toxicity levels in cancer cells, and they are involved in GNP-induced radiosensitization [20]. Of these biochemical effects, the GNPs’ effects on the radiation-induced bystander effect (RIBE) have recently received attention. RIBE is the phenomenon by which non-irradiated cells exhibit similar ionizing radiation damage as a result of signals received from nearby irradiated cells [21]. The bystander signals involved in this process may cause altered gene expression, DNA and chromosomal damage, cell proliferation alteration, or cell death in non-irradiated cells [22]. RIBEs have been demonstrated using a range of experimental systems with multiple biological endpoints; this has been the case since they were first identified by Nagasawa and Little in 1992 [23]. The underlying mechanisms mediating RIBE responses have been extensively studied and it has been shown that reactive oxygen and nitrogen species (ROS/NOS), DNA repair proteins, cytokines, ligands, extracellular DNA (ecDNA), microRNA (miRNA), and membrane molecules are the main materials that are released from targeted cells. Moreover, they are transferred to non-targeted cells via gap junction intercellular communication (GJIC) and the media/circulatory system [24].

In standard radiotherapy, where uniform fields are delivered and all cells are directly exposed to radiation, RIBE phenomena can be neglected; however, the role of RIBEs may become more influential when heterogeneous or non-uniform fields are considered. Although most clinical radiotherapy focuses on uniform exposures, there are some examples of non-uniform plans being utilized in the clinic, including spatially fractionated radiotherapy (or GRID), mini-beam radiotherapy (MBRT), microbeam radiotherapy (MRT), and dose-painting radiotherapy [25]. Healthy tissue has been shown to be more tolerant of spatially fractionated dose fields than tumor tissue, thus allowing for a high dose to be delivered in a single fraction. Spatially fractionated GRID purposely irradiates the tumor with highly non-uniform dose fields containing steep dose gradients [26,27]. Similarly to GRID therapy, MBRT and MRT are also characterized by alternating distributions of high and low doses, but on a much smaller scale (a hundred micrometers) [28]. Dose-painting radiotherapy allows for the heterogeneous delivery of high radiation doses within the tumor by targeting one or more regions of interest that are defined by functional imaging [29]. Furthermore, Monte Carlo simulation studies have shown that GNP-induced dose enhancement can be increased by a factor of 3000 compared with doses originating from a hypothetical water nanoparticle, but only at microscopic distances of 10 µm [10,11]. The heterogenous distribution of GNPs, combined with the highly localized energy depositions made by GNPs, resemble the dose pattern of spatially fractionated radiotherapy [30]; thus, RIBE plays an important role in GNP-enhanced radiation therapy.

RIBE responses have been observed in many cell types such as lymphocytes, fibroblasts, endothelial, and cancer cells [31]. Rostami et al. first investigated the effects of glucose-coated GNPs (Glu-GNPs) on the RIBE in MCF-7 (human breast cancer) and QUDB (human lung cancer) cell lines [32]. Their results demonstrated Glu-GNPs’ enhanced RIBEs in QUDB cells, but there was no RIBE enhancement in MCF-7 cells. This observation suggests that the impact of GNPs on the RIBE is cell type specific (i.e., some cell types are unable to produce bystander signals whereas others are unable to respond to bystander signals). To the best of our knowledge, there are no studies focusing upon the impact of GNP-induced radiosensitization on the RIBE in prostate cancer cells. The present study was carried out to investigate how actively targeting GNPs impacts the RIBE response in prostate cancer LNCaP cells. LNCaP cells were selected as the focus in this study because they can model a wide range of prostate cancer stages, including the transition from androgen dependence to castration resistance and bone metastasis. Moreover, the high expression of PSMA means that the LNCaP cancer model is a good choice with which to develop active-targeted nanotherapeutic strategies for prostate cancer. Anti-PSMA antibodies were conjugated with PEGylated GNPs to develop PSMA-targeted GNPs (PGNPs). A conditioned medium transfer technique was employed to evaluate the RIBE responses. The yield of the RIBE signal from irradiated cells, and the sensitivity of non-irradiated cells with regard to the RIBE signal, was used to investigate the impact of PGNPs on the RIBE in prostate cancer cells. As control groups, RIBE responses were also implemented for use on other prostate cancer cell lines (including PC3, and 22Rv1) as well on normal prostate cell lines (RWPE-1).

## 2. Materials and Methods

### 2.1. Synthesis of PSMA-Targeted GNPs (PGNPs)

Commercially available PEGylated GNPs (Creative Diagnostics, NY, USA) were used as substrates for functionalization in this study. The GNP concentration was quantified by optical absorption spectra which were determined using UV-vis spectroscopy (240 nm–780 nm, Nanodrop, Thermo Fisher Scientific, Waltham, MA, USA). To realize active and passive prostate cancer targeting, anti-PSMA and mouse IgG antibodies (Creative Diagnostics, NY, USA) were coupled with PEGylated GNPs using EDC/NHS chemistry. First, 0.2 M 1-ethyl-3-(-3-dimethylaminopropyl) carbodiimide hydrochloride (EDC) (#22980; Thermo Fisher Scientific, Waltham, MA, USA) and 0.2 M N-hydroxysuccinimide (NHS) (#24500; Thermo Fisher Scientific, Waltham, MA, USA) were simultaneously added to a solution that included OD = 1 GNPs, 0.1 mg/mL antibody, and 0.1 M sodium borate buffer. This mixture was incubated at room temperature for 24 h. Then, the conjugation solution was washed out by three centrifugations at 15,000× *g* for 30 min, and the final GNP pellet was stored in Milli-Q water (Millipore, Bedford, MA, USA) to obtain the desired concentration.

### 2.2. Characterization of PGNPs

Transmission electron microscopy was used to determine the shape and size of the developed PGNPs. PGNPs were cast onto a carbon Formvar-coated copper grid for 30 min. Excess liquid was absorbed using filter paper, and the grid was allowed to air dry overnight. The grids were viewed at 80 kV in a JEOL JEM-1400 transmission electron microscope (JEOL USA, Peabody, MA, USA) and images were captured with an AMT BioSprint digital camera (Advanced Microscopy Techniques, Woburn, MA, USA). Additional characterizations of nanoparticles have been detailed in our previously published works that focus on PGNPs [19].

### 2.3. Cell Culture

Human prostate cancer cells (LNCaP, PC3, and 22Rv1) were obtained from ATCC. Cells were cultured in RPMI1640 media (ThermoFisher Scientific, Waltham, MA, USA) with 10% fetal bovine serum (FBS, GeminiBio, Sacramento, CA, USA) and 1% penicillin-streptomycin (ThermoFisher Scientific, Waltham, MA, USA). The cells were grown in a CO_2_ incubator at 37 °C and subcultured when they reached 80–90% confluency. Normal prostate cells (RWPE-1) were cultured in the same conditions, although keratinocyte serum-free media (ThermoFisher Scientific, Waltham, MA, USA) was used instead of RPMI1640.

### 2.4. PGNP Cytotoxicity

A MTT viability assay was used to assess the cytotoxicity of treatments on LNCaP cells. Moreover, 24 h after the treatments, LNCaP cells in 96-well plates were stained with 1 M yellow tetrazolium substrate (3-(4,5-dimethylthiazol-2-yl)-2, 5-diphenyltetrazolium bromide) (MTT, Thermo Fisher Scientific, Waltham, MA, USA) in serum-free RPMI1640, for 4 h, at 37 °C. The supernatant was removed from the wells and formazan crystals were dissolved with the addition of 100 µL DMSO. This was added to each well and left for 30 min at 37 °C. An additional column of empty wells received 100µL of DMSO, which were to be read as blanks. Microplates were read on a Clariostar microplate (BMG Labtech, Ortenberg, Germany) reader for absorbance at 570 nm. MTT assay results were normalized to ensure no treatment absorptions.

### 2.5. Cellular PGNP Uptake Assay

LNCaP cells were treated with 50, 100, 150, 250, 500, and 750 μg/mL PGNPs in serum-free RPMI1640 media for 24 h. Serum-free media were used to avoid changes to the nanoparticle surface as a result of FBS proteins. After 24 h, the cells were washed three times with PBS, and they were replenished with fresh media supplemented with FBS and pen-strep prior to running assays. PGNP concentrations were measured using UV-Vis spectrophotometry on a Nanodrop Spectrophotometer before and after treatment. Spectrophotometry was conducted using a range between 280 nm to 700 nm, where the peak intensity at 520 nm was used to quantify PGNP concentration; this was achieved by benchmarking against the manufacturer’s recorded GNP optical density. The total cellular uptake of PGNP was calculated as the PGNP’s concentration difference in media before and after 24 h of treatment. Cells plated on coverslips were used to identify the intracellular distribution of PGNPs with TEM. PGNP-treated cells were fixed overnight in 2% glutaraldehyde, in 0.1 M phosphate buffer; then, they were post-fixed for 1 h in 2% osmium tetroxide in 0.1 M phosphate buffer, dehydrated through a series of graded ethanols, and embedded in an EM-bed (Electron Microscopy Sciences, Fort Washington, PA, USA). The glass coverslip was dissolved in hydrofluoric acid. In addition, 100 nm sections were cut on a Leica Ultracut EM UC7 ultramicrotome and stained with uranyl acetate and lead citrate. Images were captured on the same setup as the PGNPs on grids.

### 2.6. RIBE on LINCaP Cells

Conditioned media was collected from LNCaP cells to induce RIBE in LNCaP bystander cells. Prior to irradiation, cells were washed three times with PBS, and serum-free RPMI1640 was added to the culture plates. Cells were irradiated between 0–8 Gy at 2.0 Gy/min on a 160 kV (0.0775 Å) Radsource RS-2000 Biological Irradiator (Rad Source Technologies, Buford, GA, USA) with cells placed at 28 cm SSD (25 cm diameter circular field size). Following irradiation, cells were incubated at 37 °C for 1 h. Media from sham and irradiated cells were harvested and filtered through a 0.45 µm polyethersulfone filter (VWR, Radnor, PA, USA). All conditioned media were prepared immediately before use.

### 2.7. Effects of PGNPs on RIBE

To assess the effect of PGNPs on the RIBE, LNCaP cells were divided into three groups; bystander cells treated with conditioned media from irradiated cells (PBS-CM), bystander cells treated with conditioned media from PGNP treated irradiated cells (PGNP-CM), and PGNP treated bystander cells treated with conditioned media (PBS-CM+PGNP).

### 2.8. Clonogenic Assay

LNCaP cells were grown in 6-well dishes, and they were treated as mentioned above. Irradiated or PBS-CM treated cells were trypsinized, then replated on 6 cm dishes with varying cell numbers. Following 3 weeks of growth, plates were washed three times with cold PBS, fixed in 4% paraformaldehyde (Thermo Fisher Scientific, Waltham, MA, USA), and stained with 0.5% crystal violet (Thermo Fisher Scientific, Waltham, MA, USA) for 30 min. Plates were washed with tap water, imaged on a Chemidoc touch (Bio-rad, Hercules, CA, USA), and counted using particle counting on ImageJ.

### 2.9. γH2AX Assay

LNCaP cells were grown in chamber slides (Thermo Fisher Scientific, Waltham, MA, USA), treated as mentioned above, and fixed in 4% neutral buffered formaldehyde, permeabilized with 0.1% Triton X-100, and blocked with 1% bovine serum albumin in PBS-tween containing 5% goat serum. Slides were incubated with an antibody in order to hosphor-H2AX (1:500, CST). This was followed by incubation with goat-anti-rabbit Alexa488 (1:2000, Invitrogen, Waltham, MA, USA) and the slides were then mounted with a Prolong gold antifade reagent with DAPI (Invitrogen, Waltham, MA, USA). Cells were analyzed on a Leica confocal microscope (Leica Microsystems, Deerfield, IL, USA) with ×63 magnification. All γH2AX foci were counted using ImageJ particle counting.

### 2.10. ROS Quantification

LNCaP cells in 96-well plates were preloaded with 20 µM of DCFH-DA (Sigma-Aldrich, St. Louis, MO, USA) in serum-free RPMI1640 for 45 min. Afterwards, cells were washed three times with PBS and fresh serum-free RPMI1640 media; alternatively, conditioned media was added. For direct irradiation ROS studies, 0–8 Gy of radiation was administered to the cells. DCFH-DA fluorescence at 480 nm excitation/535 nm emission was scanned on a Clariostar microplate reader 1 h post irradiation or post conditioned media treatment.

### 2.11. Statistical Analysis

GraphPad Prism version 9.0 software was used for all statistical analyses. Differences between experimental groups were assessed using an unpaired *t*-test during a comparison of the two groups.

## 3. Results

### 3.1. Characterization, Cell Uptake, and Cytotoxicity of PGNPs

The performance of PGNPs with respect to internalization and biocompatibility was assessed for effective usage as radiosensitizers. A TEM image shows the spherical monodispersed nanoparticles following conjugation with the PSMA-antibody, with a mean diameter of 17.1 nm ± 0.28 nm (Figure 1A,B).

Incubating PSMA-expressing LNCaP cells with the developed PGNPs revealed the internalization of PGNPs in small clusters throughout the cell (Figure 1C,D). Cellular uptake analysis (Figure 1E) demonstrated the enhanced PGNP levels at increasing concentrations, with the maximum saturation point starting from a concentration of 250 µg/mL. On average, a single LNCaP cell accumulated 12 picograms PGNP under 250 µg/mL of PGNP treatment (Figure 1E). The percentage cell viability (Figure 1F) demonstrated that there exists no statistically significant cytotoxicity for PGNPs with concentrations up to 250 µg/mL; therefore, the biocompatible concentration of the 250 µg/mL PGNPs was applied to the following RIBE experiments, which falls under the calculated IC50 value of 420 µg/mL.

To validate the specificity of active-targeting PGNPs, GNPs conjugated with mouse IgG (passive-targeting) and anti-PSMA IgG (active-targeting) were used for treatment on prostate cancer cell lines (including LNCaP, PC-3, and 22Rv1) and normal prostate endothelial cell lines (RWPE-1). First, we compared the expression of PSMA with different prostate cancer cell lines, as well as normal prostate cell lines. Western blot analysis demonstrated that LNCaP and 22Rv1 express significant levels of PSMA, whereas PC-3 and RWPE-1 are PSMA negative (Figure 1G,H). Then, we compared the targeting efficiency of PSMA-passive and -active GNPs on different cell lines by labelling GNPs with fluorophores and treating each cell line with 250 µg/mL of GNPs for 24 hrs. Compared with PSMA-passive GNPs, PSMA-active GNPs exhibited significantly increased levels of fluorescence signals in LNCaP and 22Rv1 cell lines, but not in PC3 and RWPE-1 cells; therefore, moving forward, LNCaP cells were identified as the optimal cell line with which to study the RIBE because of their high PSMA expression (Figure 1I,J).

### 3.2. Effect of Radiation Dose or PGNP Concentration on Radiosensitization

PGNP-induced radiosensitization in LNCaP cells was assessed through a clonogenic cell survival assay and γ-H2AX assays. To examine the effects of radiation doses and PGNP concentrations on in vitro radiosensitization, the LNCaP cells were incubated with PGNPs at different concentrations (0, 50, 100, 150, 250 µg/mL) and irradiated with different doses (0, 2, 4, 6, 8 Gy). Clonogenic cell survival assay results (Figure 2A) demonstrated enhanced radiosensitization levels in LNCaP cells when subjected to increased PGNP concentrations. Moreover, for a specific PGNP concentration, a higher radiosensitization level was observed at a higher radiation dose. Figure 2B compares the radiosensitization of PGNPs, using various concentrations, under 2-Gy irradiation. When treated with 150 µg/mL PGNPs or more, there was a statistically significant difference regarding the radiosensitization levels between the groups that were treated without and with PGNPs, thus suggesting that 150 µg/mL was the minimum concentration required for PGNP-induced radiosensitization under 2 Gy radiation.

γ-H2AX foci were assessed as biomarkers for double-strand DNA damages. As shown by the fluorescence images in Figure 2C, γ-H2AX foci could be clearly distinguished after the irradiation (2 Gy) of LNCaP cells incubated with/without PGNPs (250 µg/mL). Additionally, the non-irradiated LNCaP cells that were incubated with 250 µg/mL PGNPs showed no signs of foci formation, thus showing that these nanoparticles have no DNA damage effects without radiation. The average number of γ-H2AX foci per cell was counted during 2-Gy irradiation at different PGNP concentrations (0, 50, 100, 150, 250 µg/mL), and the results are presented in Figure 2D. PGNP treatment showed a dose dependent increase in foci count with an increase in GNP concentration (Figure 2D). A statistically significant increase in foci count was seen at 250 µg/mL.

### 3.3. Effect of PGNP on RIBE Signaling Intensity

The effect of PGNPs on RIBE signaling was assessed through the conditioned medium (CM) transfer procedure (Figure 3A), where the bystander LNCaP cells were treated with PBS- or PGNP-CM that was extracted from irradiated cells that were pretreated with PBS or 250 µg/mL PGNPs. Figure 3B shows the radiation response of PBS-CM treatments on LNCaP/PC3/22Rv1/RWPE-1 bystander cells. The RIBE response of the bystander cells treated with the PBS-CM from the irradiation groups (2, 4, 6, 8 Gy) were statistically different from the non-irradiation group (*p* < 0.05); this indicated the radiation dose dependence of the RIBE in LNCaP cells. This dose-dependence effect was also observed in PC-3 and 22Rv1 cells, but not in RWPE-1 cells. Moreover, 22Rv1 cells showed a weaker RIBE response than LNCaP/PC3 prostate cancer cell lines. MTT assays (Figure 3C) showed the statistically different cell viability between LNCaP bystander cells treated with 2 Gy-PGNP-CM and 2 Gy-PBS-CM. This aligns with the PGNP-enhanced radiosensitization (as shown in Figure 2) and the radiation dose dependence of the RIBE (as shown in Figure 3B). γ-H2AX assays (Figure 3D) demonstrated the significantly different number of foci, or the DNA damage between bystander cells, treated with 2 Gy-PGNP-CM and 2 Gy-PBS-CM. Furthermore, the clonogenic cell survival assay showed significantly decreased colony formations of bystander cells treated with 2 Gy-PGNP-CM compared with the 2 Gy-PBS-CM group. (Figure 3E); therefore, PGNP-induced radiosensitization further enhanced the RIBE signaling in LNCaP cells. In Figure 3C–E, no significant differences were observed between the LNCaP bystander cells treated with 0 Gy-PGNP-CM or 0 Gy-PBS-CM, which further illustrates the non-cytotoxicity of the 250 µg/mL PGNPs.

### 3.4. Effect of PGNP on the Sensitivity of Bystander Cells to the RIBE

The influence of PGNPs on the sensitivity of bystander cells to the RIBE was assessed on the PBS-CM treated LNCaP cells that were pretreated with or without 250µg/mL PGNPs. The detailed workflow is outlined in Figure 4A. Moreover, 2 Gy irradiation was adopted here because 2 Gy was the minimum RT dose that enabled the generation of a significant RIBE response in LNCaP bystander cells, as shown in Figure 3B. The RIBE response was evaluated in the following four groups: 0 Gy-PBS-CM on no-PGNPs-loaded cells (0 Gy-PBS-CM), 0 Gy-PBS-CM treatment on PGNPs-loaded bystander cells (0 Gy-PBS-CM+PGNPs), 2 Gy-PBS-CM on no-PGNPs-loaded cells (2 Gy-PBS-CM+PGNPs), and 2 Gy-PBS-CM on PGNPs-loaded cells (2 Gy-PBS-CM+PGNPs). MTT assays in Figure 4B demonstrated that there was significantly lower cell viability in the 2 Gy-PBS-CM+PGNPs group (85%) compared with the 2 Gy-PBS-CM group (92%). Additionally, compared with the 2 Gy-PBS-CM treatment group, the 2 Gy-PBS-CM+PGNPs group showed both an increased foci count in the γH2AX assay and decreased colony formation in the clonogenic assay (Figure 4C,D), thus indicating the PGNP-boosted susceptibility of LNCaP bystander cells to RIBE damage.

### 3.5. Effect of PGNP on ROS Production

As a common and potent type of RIBE mediator, ROS was measured with DCFDA in order to study the impact of PGNPs on ROS production in relation to the RIBE response. Figure 5A shows the percentage of ROS change between LNCaP cells treated with PGNPs at various nontoxic concentrations (10, 50, 100, 150, 200, 250 µg/mL). There were no significant differences between the different treatment groups, thus indicating that no clear relationship between PGNP concentration and ROS production exists. Figure 5B compares the intracellular ROS levels of irradiated LNCaP cells that were pretreated with and without PGNPs (250 µg/mL). As shown, irradiation-induced ROS production that was approximately 1.2× higher at every radiation dose (Figure 5B) in LNCaP cells pretreated with PGNPs, compared with LNCaP cells pretreated without PGNPs; this indicates that PGNPs boosted ROS production during radiation treatment. Figure 5C exhibits the effect of PGNPs on ROS production in LNCaP bystander cells in the following groups: treated with PBS-CM, treated with PGNP-CM, incubated with PGNPs for 24 h, and treated with PBS-CM (PBS-CM+PGNPs). For each treatment group, the bystander cells treated with conditioned media harvested from irradiated cells that were subjected to higher doses produced more ROS, which highlights the radiation dose dependence of ROS production with regard to the RIBE. Due to the GNP-induced radiosensitization, the bystander cells treated with PGNP-CM showed a larger ROS fold change, compared with the cells treated with PBS-CM. Compared with the PBS- and PGNP-CM treatment groups, the PBS-CM+PGNP group demonstrated a much higher level of ROS production, thus indicating the effect of internalized PGNPs on the sensitivity of bystander cells to the RIBE response. 

## 4. Discussion

Actively targeting tumors is crucial for GNP-aided radiosensitization; indeed, it enhances tumor specific accumulation and cell killing while sparing surrounding healthy tissue. In this study, internalization analysis on TEM images demonstrated the intracellular biodistribution of the developed PGNPs in cell cytoplasm. Although nuclear localization would theoretically induce a greater number of DNA double strand breaks through increased secondary electron scatter around GNPs, recent studies have suggested mitochondrial damage poses an additional threat to long-term cancer cell proliferation [33]; therefore, the cytoplasmic effects of radiosensitization are equally valuable in terms of direct DNA damage. Furthermore, uptake experiments indicated that cellular GNP saturation occurred at approximately 250 µg/mL, with minimal toxicity occurring within the LNCaP cells, and accumulations reaching 12 pg/cell. This value indicates lower toxicity levels than other studies performed with GNPs and LNCaP cells, which reported IC50 values between 100–200 µg/mL after 24 h [34,35,36]. This difference is mainly because of the media used during incubation with GNPs. In other studies, complete growth media were used, which allows the control group to continue growing; however, in our study, serum free media was used, which significantly reduces cell proliferation in the control cells. GNPs are known to induce cell cycle arrest; therefore, viability studies using complete media generate an IC50 based on a control cell line with cell proliferation, whereas GNP treated cells can become senescent [37]. However, this study employs serum-free media, which ensures that the MTT assay specifically measures cell death. This toxicity and uptake data agree with previous targeted nanoparticle studies [38]. Furthermore, this study confirmed that the anti-PSMA antibody functionalized PSMA-targeting GNPs are effective radiosensitizers for prostate cancer cells through a clonogenic assay and γ-H2AX assay. Both the clonogenic assay and γ-H2AX results showed a GNP dose dependent response to LNCaP cell radiosensitivity. The focus of the LNCaP prostate cancer cell line in the present study was motivated by the high expression of PSMA in LNCaP cells, and studies showing the high sensitivity of LNCaP cells to radiotherapy. Western blot confirmed the highest expression of PSMA in LNCaP cells compared with other prostate cancer cell lines (including PC3 and 22Rv1), as well as a normal prostate cell line (i.e., RWPE-1). Both LNCaP and 22Rv1 cell lines demonstrated an elevated uptake of PGNPs. Interestingly, the normal prostate cells (i.e., RWPE-1) demonstrated a high uptake of the GNPs used in this study, under the same in vitro incubation conditions as the prostate cancer cell groups. During GNP-assisted radiotherapy, tumor specificity occurs because 5–150 nm-sized GNPs accumulate in the tumor tissues as a result of the enhanced permeation and retention (EPR) effect; however, in contrast to the in vitro scenario, normal prostate tissues accumulate significantly fewer GNPs than prostate tumors in vivo due to their low levels of EPR. Therefore, in the in vivo scenario, normal prostate cells only accumulate a small number of GNPs, even though they do show a high uptake of GNPs in vitro.

Moreover, GNPs affected the RIBE in LNCaP cells. LNCaP bystander cell viability decreased with increased radiation doses up to 8 Gy when using the media transfer technique. The dose dependence of the RIBE response was also observed in other bystander prostate cancer cells (PC3, 22Rv1), but not in normal prostate cells (RWPE-1). As expected, radioresistant 22Rv1 cells showed a minimal response to the RIBE compared with LNCaP and PC3 cells. Additionally, PGNP-CM treatment induced increased cell death in bystander cells compared with bystander cells treated with PBS-CM. The increased cell death in bystander cells after PGNP-CM treatment suggests that the GNPs increased the radiation damage in irradiated cells, and thereby increased RIBE signaling because the RIBE in LNCaP cells was dose dependent for a wide range of cells. Previous studies on breast cancer cells showed that the dose dependence of cell lines in relation to the RIBE is critical to a GNP-enhanced RIBE [32]. γ-H2AX and clonogenic cell survival assays further confirmed the radiosensitization of LNCaP cells through the RIBE. This study only investigated the effects of the cytotoxicity of active-targeting PGNPs on LNCaP cells. Future relevant studies are required to investigate the effects of the cytotoxicity of PGNPs on PC3, 22Rv1, and RWPE-1 cell lines, which would enable an exploration of the impact of PGNPs on different prostate cancer cells as well as normal prostate cells.

PGNPs also increased the LNCaP cells’ sensitivity to RIBE signals in a cell viability assay, γ-H2AX, and a clonogenic assay. The increased total cellular ROS following conditioned media exposure in PGNP-treated bystander cells demonstrated that imbalanced cellular redox potential plays a role. Previous studies indicate that metal nanoparticle treatment augments antioxidant and ROS-generating enzyme levels and leads to increased cellular ROS production [39]. Culcasi et al. and Wilhelmi et al. identified an upregulation in ROS-generating NADPH oxidase (NOX) enzymes that played a key role in metal nanoparticle cytotoxicity through increased cellular ROS, which thus led to DNA fragmentation [40]. Enzymes such as NOX enzymes, which can induce oxidative stress, play a key role in radiation therapy and the RIBE because studies show that the inhibition of NOX and other ROS-generating enzymes can significantly reduce DNA damage [41,42]. Furthermore, ROS clearance through enzymes such as superoxide dismutase (SOD), glutathione peroxidase (GPx), and catalases maintain cellular redox potential. A variety of metal nanoparticle studies in various tissues demonstrate a decrease in the expression of these enzymes, including a decrease in GPx in TiO_2_, a decrease in SOD and catalases in CuO, and a decrease in SOD in iron oxide [39]. Although there is a dearth of experiments with regard to GNPs and antioxidant enzyme changes, it is a potential parameter requiring further investigation; therefore, the dose dependent increase in total cellular ROS upon the irradiation of PGNP-treated cells likely occurs as a result of a two-fold effect. First, the physical dose enhancement of PGNPs increased local dose deposition, leading to physical radiation dose dependent ROS production, and PGNP-impaired cellular redox maintenance with biologically boosted ROS levels. The latter was observed in bystander cells with conditioned media transfer studies. Ultimately, the elevated ROS levels occurred as a result of significantly increased enzymatic ROS production, which produced a surplus, and the conditioned media treatment in the bystander cells, which is likely to be the reason for the increased DNA damage seen in the γ-H2AX assay; this is also likely to be the case for the increased radiosensitization seen in the clonogenic survival assays.

A key element of this study was the use of conditioned media as a tool for inducing RIBE. Conditioned media utilizes soluble cytokines such as IL-1, IL-2, IL-6, IL-8, TNF-alpha, and TGF-beta [43]. Many of these cytokines are highly expressed in immune cells, such as macrophages, but in this study, we only studied the soluble factors in conditioned media produced by LNCaP cells, and we treated them to LNCaP cells; situations such as these can only be observed locally within the tumor. Therefore, the impact of the RIBE in distal regions or RIBE signals from immune cells could augment the observed effect. Furthermore, using the conditioned media technique caused the loss of two major RIBE signals: free radicals and gap junction signals [24]. Most free radicals are eliminated quickly before the conditioned media is transferred. Additionally, the strongest known signaling of the RIBE occurs through gap junctions. Previous RIBE studies have utilized partial culture irradiation, whereby a few cells in a single culture dish are radiated [44]; however, this RIBE is very localized, with an unclear, limited range [45]. Therefore, RIBE enhancement with PGNPs may have a different effect if studied with the partial irradiation method. Furthermore, recent studies focus on the impact of exosomes on the radiation-induced bystander effect [46,47]. A major impact of filtering the conditioned media with 0.45 µm sterile filter is the preservation of exosomes in the conditioned media. Lastly, the use of conditioned media provided in vivo translation opportunities for this study. The injection of conditioned media into mice could potentially show tumor treatment effects, and vice versa, the serum from irradiated mice can be used in vitro. These studies reveal the degree and range of RIBE signaling changes that GNPs can affect. The results presented here suggest that the bystander prostate cancer cells would experience greater cell death rates when these cells are enriched with GNPs and treated with RIBE signals from irradiated cancer cells with internalized GNPs.

A potential caveat of this study is the fact that the RIBE enhancement of cancer cells treated with kV photons is limited to LNCaP prostate cancer cells. Previous studies have demonstrated that various cells have different responses to the RIBE [46]. For instance, some cells (such as the breast cancer cell line, MCF7) are known to show a radiation-induced bystander effect but not in a dose dependent manner. On the other hand, QUDB cells have a dose dependent response [32]. Another study showed that there is a radiation-induced cell growth inhibitory bystander response, but only at high radiation doses, not low radiation does [48]. This is particularly interesting as LNCaP demonstrated a consistent RIBE, including low and high radiation doses. To consider the full effect of the radiation-induced signaling of the RIBE in prostate cancer cells, studies into other prostate cancer cells, as well as normal tissue prostate fibroblast cells, can provide an understanding of how the RIBE could be used in a clinical setting. Additionally, they could also demonstrate the effect of GNPs on these bystander effects. Moreover, RIBEs have been known to differ based on their radiation quality and LET. The RIBE has been studied using various radiation sources, including photons, particles, and heavy ions at varying degrees of intensity [49,50,51]. Although many of these have not been compared against each other, several studies indicate that the magnitude of bystander effect is dependent on LET [52]. Changes between radiation sources have been shown to change dose dependence as well as relieve the RIBE burden entirely [53]. This is particularly important when studying GNP-induced radiosensitization because the mechanism of GNP-mediated radiosensitization is different depending on the radiation source. For instance, interactions between kV energy beams and GNPs leads to the emission of short-range secondary electrons and radiosensitization. The use of kV energies in this study maximizes the effect of secondary electron scatter compared with other modalities [8]. Moreover, MV irradiated GNPs induce radiosensitization through increased mitochondrial toxicity, and Monte Carlo and in vitro studies suggest protons or heavy ions generate a massive increase in ROS production [54]. All these modalities are currently used clinically for prostate cancer treatments in the form of brachytherapy or external beam radiotherapy, and the 2 Gy increments match the conventional fractionation schemes of standard treatment schedules. As our study suggests, the RIBE contributes to GNP-mediated radiosensitivity through ROS production, and the increase in sensitivity may be further enhanced when protons and heavy ion-irradiated cells are the donors for the conditioned media. Lastly, this study focuses on GNPs, but a multitude of metal nanoparticles have demonstrated radiosensitization capabilities. A diverse group of these metal nanoparticles induce cellular redox changes, which suggest that the effects observed in this study are not limited to GNPs. In addition, several studies indicate that metal nanoparticles introduce cellular toxicity through ion shedding heavy metal poisoning with TiO2, carcinogenesis from ZnO, and inflammation from impurities in Ag [55,56,57]. GNPs are free of these problems, but these toxicities relate largely to ROS generation from the nanoparticles, which could further impact RIBE sensitivity in a similar manner to GNPs.

The clinical benefit of the bystander effect can be observed through a series of in vivo experiments, including the use of spatially fractionated radiotherapy [25]. Spatially fractionated radiotherapy is the delivery of a radiation dose to smaller fields without delivering radiation to the entire tumor to reduce toxicity. Preclinical studies in mouse models demonstrated that a 10 Gy dose to a small region of the tumor would cause radiation damage to the surrounding tumor tissue [58]. Furthermore, this bystander effect showed changes in gene expression for DNA repair, cell cycle arrest, and apoptosis, as is the case with the changes seen in irradiated tumor cells. Although studies on spatially fractionated radiotherapy have been conducted since the 1950s, inventions of multi-leaf collimators have recently shown promising results regarding prostate tumor responses [59]. The implication of enhanced bystander effects from GNPs suggests that they can be valuable in combination with spatially fractionated radiotherapy to enhance the bystander effect. The results presented here suggest that spatially fractioned radiotherapy would benefit from GNPs, even if GNPs are not homogenously distributed throughout the tumor. Factors such as hypoxia, nanoparticle targeting agents, and tumor microenvironments can affect the distribution of GNPs, but a spatially fractionated study would enhance tumor treatment efficacy, regardless of the GNP content in each beamlet. Furthermore, it is possible to use imaging studies to target regions with high GNP concentrations in order to selectively utilize the effects that GNPs have on the RIBE for dose painting. In vivo experiments concerning GNPs, combined with spatially fractionated radiotherapy, could also reveal the extent of the GNP-enhanced RIBE to the surrounding tissue. An additional weakness in this study, for clinical reference, is the use of a single radiation dose to induce a bystander effect. For clinical use, the bystander effect would require fractionated radiation. It is important to note that fractionated radiation doses have had mixed results in RIBE studies [60]; however, previous studies on the pharmacodynamics of gold nanoparticles indicate that fractionated radiation therapy is completely feasible for GNP-enhanced radiation therapies [19].

Finally, this study suggested that the radiosensitization effects of off-targeted GNPs could be a potential double-edged sword. The increased effect of RIBE sensitivity as a result of nanoparticles could induce a greater number of secondary tumorigeneses, as nanoparticles are known to accumulate in various organs such as the liver, spleen, and kidneys [33,61]. These effects can be particularly worrisome since secondary cancers in the lung are prevalent with prostate cancer treatments [62]. Previous studies have demonstrated low accumulations of GNPs in the lung, which may therefore mitigate this risk, but the high accumulations in the liver, kidney, and spleen may increase the risk of secondary cancers in these organs when treated with GNPs. This form of distant organ damage may be described more specifically as an ‘abscopal effect’, and it is not particular to the bystander effect; however, the proper control of nanoparticle biodistribution with targeted nanoparticles could be critical for GNP radiotherapy safety.

Studies on the RIBE present further biological explanations on the variances observed between in silico GNP studies and in vivo/in vitro experiments. Several Monte Carlo studies have demonstrated that GNPs can increase local dose deposition by up to 200× per nanoparticle with 250 kvP photon beams [63]; however, the increased secondary electron scatter fails to account for the total radiosensitization observed in vitro, especially for megavoltage energies [20]. Studies show that GNP radiosensitization can occur due to a variety of reasons, including physical, chemical, and biological [20]. Physically, GNPs increase the secondary electron scatter, and chemically, the GNPs form free radicals to make DNA more susceptible to radiation damage. A myriad of biological effects occur, including cell cycle changes, mitochondrial damage, and DNA repair inhibition. Studies in glucose-capped GNPs increased the cell population in the G2/M phase, which is the most radiosensitive phase. Secondly, several studies on the GNPs’ impact on mitochondria show radiosensitization through the loss of mitochondrial potential, thus leading to necrosis, caspase-related mitochondrial dysfunction, and mitochondrial membrane polarization; this leads to apoptosis when combined with radiation. Thirdly, H2AX studies have shown complications with DNA repair when GNPs are present, based on increased double stranded DNA damage to the foci after 24 h. RIBE changes from GNPs add to the list of biological changes which lead to increased radiosensitization, and the biological factors affecting radiosensitization from direct radiation further enhance the RIBE. Nanoparticles have recently garnered attention in radiotherapy as valuable agents for imaging and radiosensitization. As such, the direct effects of metallic nanoparticle radiosensitization and toxicity have been heavily studied [38,64]; however, the nanoparticles’ impact on the RIBE has not been well documented. The radiation-induced bystander effect can have a critical role in improving the treatment efficacy of treatment modalities, such as spatially fractionated radiotherapy, and it can also have detrimental effects in the form of off-target secondary cancers. This study introduced the potential for combining GNPs to augment the effects of the RIBE in prostate cancer treatment.

## 5. Conclusions

Prostate cancer-targeting GNPs were used to radiosensitize LNCaP prostate cancer cells. This study elucidated the impact of the bystander effect on the biological radiosensitization of prostate cancer cells with targeted GNPs. Additionally, this study highlights the increased signaling intensity of the bystander effect from GNPs in the irradiated cells, as well as the greater sensitivity to the bystander effect in non-irradiated cells as a result of GNP treatment. A plethora of studies have demonstrated that radiosensitization from metal nanoparticles occurs as a result of a multitude of biological and chemical factors; these factors impact a greater area than the originally proposed site for local dose deposition. The findings from this study contribute to knowledge concerning another biological component of GNP radiosensitization. Ultimately, these findings suggest further research should be undertaken to investigate the GNP-enhanced bystander effects, abscopal effects, distant tissue toxicity risks, and combinations of GNPs and heterogenous dose distribution treatments.

## Figures and Tables

**Figure 1 nanomaterials-12-04440-f001:**
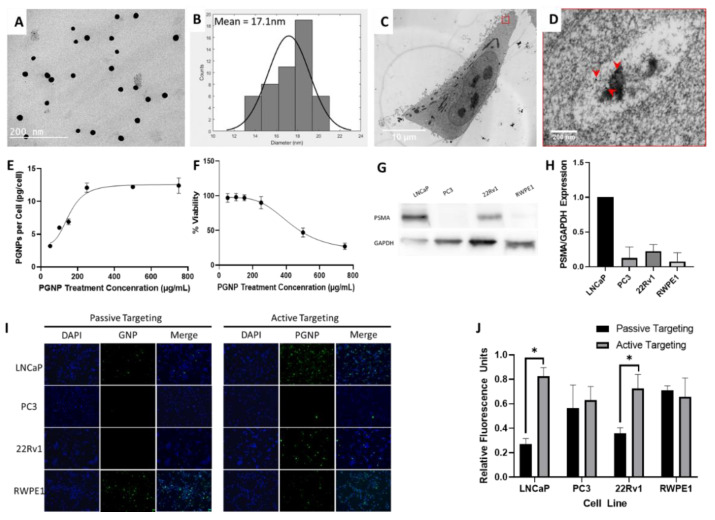
(**A**) TEM image of PSMA-antibody conjugated gold nanoparticles (PGNPs). (**B**) Histogram of PGNP diameters (mean = 17.1 nm ± 0.28 nm) from TEM images. (**C**) Representative TEM image of a LNCaP cell treated with 250 µg/mL of PGNPs. (**D**) Magnified region with arrow heads indicating internalized PGNPs. (**E**) PGNP uptake in LNCaP cells after 24 h of treatment at various PGNP doses (50 µg/mL, 100 µg/mL, 150 µg/mL, 250 µg/mL, 500 µg/mL, 750 µg/mL). (**F**) Percentage viability of the LNCaP cells incubated with different concentrations of PGNP (50 µg/mL, 100 µg/mL, 150 µg/mL, 250 µg/mL, 500 µg/mL, 750 µg/mL). (**G**) Western blot for PSMA expression in whole cell lysate of LNCaP, PC3, 22Rv1 (prostate cancer), and RWPE1 (normal prostate) cells. (**H**) Quantification of PSMA expression in the western blot normalized to a GAPDH signal. (**I**) Fluorescence microscopy image of LNCaP, PC3, 22Rv1, and RWPE1 cells treated with 250 µg/mL of passive targeting (mouse IgG conjugated) GNPs and active targeting (PSMA-antibody conjugated) PGNPs. (**J**) Quantification of fluorescence microscopy normalized to the highest signal in each cell line. * Denotes *p* < 0.05 from Welch’s *t*-test.

**Figure 2 nanomaterials-12-04440-f002:**
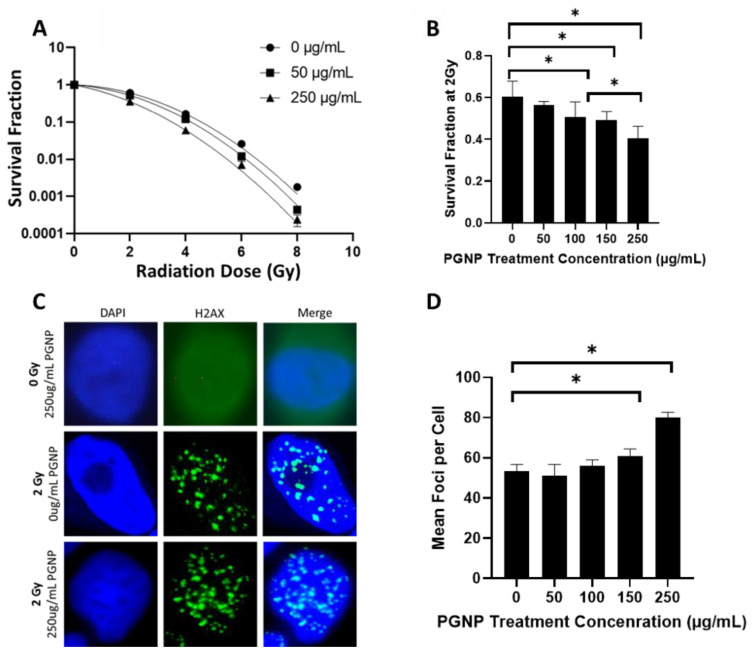
(**A**) Representative survival curve for LNCaP cells irradiated with various doses (0, 2, 4, 6, 8 Gy) following a 24 h pretreatment of PGNPs at different concentrations. (**B**) Survival fraction of 2 Gy-irradiated LNCaP cells, 24 h after the treatment with PGNPs at different concentrations. (**C**) γ-H2AX staining of LNCaP cells pretreated with 0 or 250 µg/mL PGNPs and irradiated with 0 Gy or 2 Gy. (**D**) Comparison of γ-H2AX foci counts for groups with different PGNP concentrations after 2Gy irradiation. * Denotes *p* < 0.05 from Welch’s *t*-test.

**Figure 3 nanomaterials-12-04440-f003:**
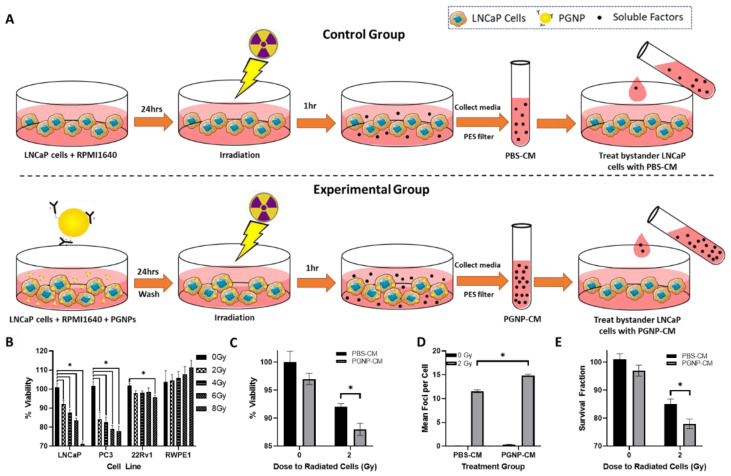
(**A**) Workflow of the conditioned medium transfer procedure for a RIBE response investigation. (**B**) MTT cell viability assays of LNCaP/PC3/22Rv1 (prostate cancer) and RWPE1 (normal prostate) bystander cells treated with PBS-CM extracted from their respective irradiated cell line cells under different radiation doses. (**C**–**E**) MTT cell viability, γH2AX, and clonogenic assays of bystander cells treated with 0 Gy-PBS-CM, 2 Gy-PBS-CM, 0 Gy-PGNP-CM, and 2 Gy-PGNP-CM. * Denotes *p* < 0.05 from Welch’s *t*-test.

**Figure 4 nanomaterials-12-04440-f004:**
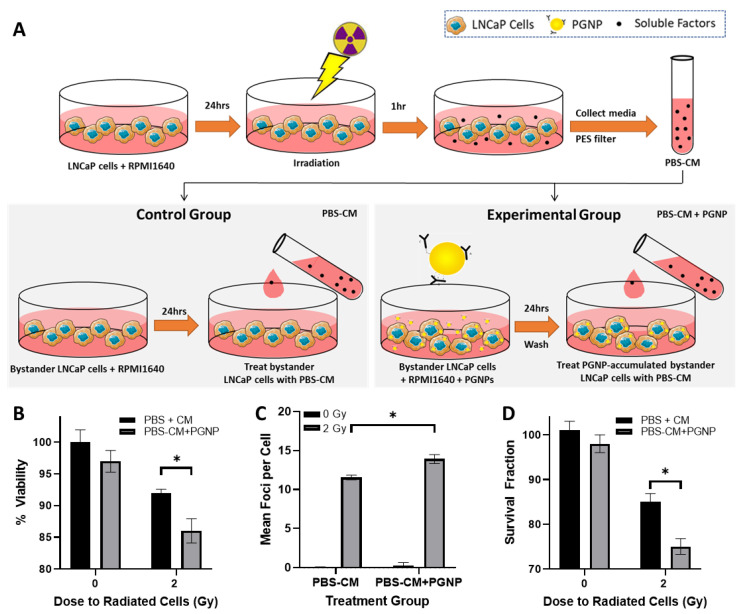
(**A**) Workflow of the conditioned medium transfer procedure in order to investigate the effect of PGNPs on the susceptibility of LNCaP bystander cells to RIBE damage. (**B**–**D**) MTT cell viability, γH2AX, and clonogenic assays of bystander cells under different treatments: 0 Gy-PBS-CM: 0 Gy-PBS-CM on bystander cells pretreated without PGNPs; 0 Gy-PBS-CM+PGNPs: 0 Gy-PBS-CM treatment on bystander cells pretreated with 250 µg/mL PGNPs; 2 Gy-PBS-CM+PGNPs: 2 Gy-PBS-CM on bystander cells pretreated without PGNPs; 2 Gy-PBS-CM+PGNPs: 2 Gy-PBS-CM on bystander cells pretreated with 250 µg/mL PGNPs. * Denotes *p* < 0.05 from Welch’s *t*-test.

**Figure 5 nanomaterials-12-04440-f005:**
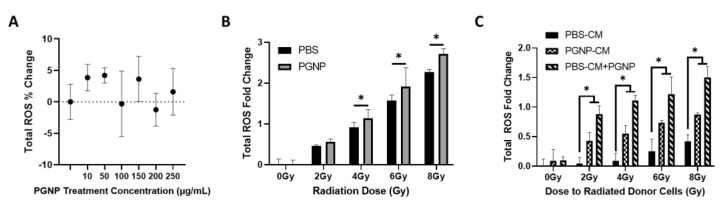
DCFDA total ROS production measurements normalized in accordance with the measurements of the untreated group. (**A**) Percentage of ROS change in LNCaP cells incubated with PGNPs under various concentrations. (**B**) Total ROS change induced by irradiation in LNCaP cells pretreated with and without 250 µg/mL PGNPs (*p* < 0.05 at 4, 6, and 8 Gy). (**C**) Total ROS change in bystander LNCaP cells in the following groups: treated with PBS-CM, treated with PGNP-CM, and bystander cells incubated with PGNPs for 24 h and treated with PBS-CM (PBS-CM+PGNPs) (*p* < 0.05 at 2, 4, 6, and 8 Gy). * Denotes *p* < 0.05 from Welch’s *t*-test.

## Data Availability

Not applicable.

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
