# Peer review of "Boosted Radiation Bystander Effect of PSMA-Targeted Gold Nanoparticles in Prostate Cancer Radiosensitization"

_nanomaterials, 2022, doi:10.3390/nano12244440_

Round 1

Reviewer 1 Report

This manuscript aims to study the effect of active-targeting GNPs on radiation-induced bystander effect (RIBE) in prostate cancer cells LNCaP, with the anti-prostate-specific membrane antigen (PSMA) antibody conjugated onto the PEGylated GNPs. In general this article is well written and the following issues should be addressed adequately to improve quality of this manuscript:

1.        Only the prostate cancer cells LNCaP are used in this study, which should be inadequate. It would be better for the authors to use normal prostate cells as well for comparison of the radiosensitization effect.

2.        It has been demonstrated that the impact of GNPs on RIBE is cell type specific, i.e. some cell types are unable to produce bystander signals while others are unable to respond to bystander signals. How about some other types of prostate cancer cells such as PC3? The authors should comment on this issue.

3.        For the LNCaP cell viability incubated with different concentrations of PGNPs, the IC50 should be calculated for comparison with that reported in the literature. It seems the IC50 is quite high in inhibiting LNCaP cell proliferation. The authors should comment on this issue.

4.        It is a pity that no animal experiment is performed to verify the radiosensitization effect observed in this in vitro study. The possible in vivo effect should be discussed.

5.        In addition to ROS measurement, the determination of antioxidant enzyme activities such as catalase, superoxide dismutase and glutathione peroxidase as well as lipid oxidation products such as MDA can be carried out. Also, the possible alteration of cell cycle distribution, mitochondrial damage and change of apoptosis or necrosis of prostate cancer cells can be discussed in more detail.

Reviewer 2 Report

The article presents the results of evaluating the effectiveness of the use of gold nanoparticles as radiosensitizers for a local increase in the radiation dose in cancer target cells. In general, the presented work is quite interesting and promising, and the results obtained can be further used for practical application, as well as understanding the fundamental principles of the processes of interaction between nanoparticles and living cells. This article has a good structure, and also corresponds to the subject of the declared journal. However, before accepting it for publication, the authors should give a number of answers to the reviewer's questions, as well as make corrections to the text of the article. An article can be recommended for publication only if the authors answer all questions.

1. To begin with, the authors should disclose in the abstract in more detail about the prospects of this study in modern materials science and the biological application of nanoparticles, as well as the choice of gold nanoparticles as the main direction on the radiation-induced effect of the bystander when interacting with cells. Why was this cell type chosen?

2. The authors should, in addition to the average particle size in Figure 1, also present the error value, as well as the degree of particle size homogeneity, in order to understand that the particles they propose have the same morphology.

3. Also, the authors should provide explanations about the details of the choice of X-rays, radiation doses, as well as the wavelength and time dependence of the set of doses. You should also pay attention to the fact that the doses have limitations, with which this was due

4. In conclusion, more details should be given about further research prospects, as well as the possibilities of using other types of particles in this direction. Also, the authors should provide a comparative analysis with other types of particles in the field of toxicity and biomedical applications.

5. When characterizing the samples, the authors should also pay attention to the structural characteristics of the samples and the degree of their resistance to the media in which they are used.

Round 2

Reviewer 1 Report

The authors have satisfactorily addressed all the comments raised by reviewers and therefore I recommend acceptance of this article for publication in Nanomaterials.